# Assessment of the Effect of Treating ‘GiSelA 5’ Softwood Cuttings with Biostimulants and Synthetic Auxin on Their Root Formation and Some of Their Physiological Parameters

**DOI:** 10.3390/plants12030658

**Published:** 2023-02-02

**Authors:** Sławomir Świerczyński

**Affiliations:** Department of Ornamental Plants, Dendrology and Pomology, Poznan University of Life Sciences, Dąbrowskiego 159, 60-594 Poznan, Poland; slawomir.swierczynski@up.poznan.pl; Tel.: +48-61848-7955

**Keywords:** sweet cherry, ‘GiSelA 5’ rootstock, propagation, biostimulants, substrate, time collection of cuttings

## Abstract

The ‘GiSelA 5’ rootstock is of high importance for growing sweet cherries because it significantly reduces the growth vigour of the trees and accelerates their fruiting. However, the method of its propagation using ‘in vitro’ cultures is expensive, prompting researchers to look for alternative methods of propagation. One of these is the root formation in shoot cuttings. The experiment presented in this paper compared the use of powdered synthetic auxins (Rhizopon AA, Ukorzeniacz AB) and their alcoholic solution (IAA, IBA), and the biostimulants Goteo and Bispeed as foliar sprays for root formation in softwood cuttings ‘GiSelA 5’ and measured some parameters of physiological processes. In addition, two different substrates of river sand with peat (1:2) and peat substrate with perlite (2:1) were used. Cuttings were obtained on two dates, in the first and third years in the month of June. Biostimulants increased the number of rooted cuttings (Goteo—16.1%, Bispeed—18.1%) without improving their growth and the intensity of most of the analyzed physiological processes compared to the control. Synthetic preparations also increased the percentage of rooted cuttings Rhizopon AA (24.4%), Ukorzeniacz AB (21.4%), auxin IBA (19.7%) and auxin IAA (14.7%), while simultaneously improving their growth and level of vital processes compared to the controls and biostimulants. The substrate consisting of peat with sand improved root formation by 6.2%, without significantly changing the growth parameters and vital functions of the cuttings. The earlier date of propagation increased the root formation percentage only by 4% and the number of roots by 14% of the ‘GiSelA 5’ rootstock cuttings.

## 1. Introduction

Plants belonging to the genus *Prunus* are classified as species that are difficult to propagate [1,2]. This is especially true for those derived from interspecies crosses [3]. One of them is the rootstocks ‘GiSelA 5’. According to some researchers [4,5], semi-dwarf rootstocks, which include ‘GiSelA 5’, perform best in orchards with a high tree planting density, in combination with strongly growing sweet cherry varieties. It enables the formation of the spindle crown of trees, the most suitable for dense plantings, and the performance of agrotechnical treatments and fruit harvesting directly from the ground. However, they require fertile soils and suitable climatic conditions, which is why it is discouraged to plant them on shallow and low-fertility soils. The process of root initiation in shoot cuttings is influenced by many factors, among them the content of phytohormones, especially auxins [6,7]. Stefančič et al. [8] state that, in the case of the ‘GiSelA 5’ rootstock, this may be due to the low amount of natural auxins accumulated in the shoots. The level of root formation in cuttings also depends on a number of factors including the age of parent plants and their etiolation before obtaining cuttings [9]. The rootstock that was analyzed in this study can be obtained most effectively through the ‘in vitro’ culture method. However, this results in a high price for the maiden sweet cherry trees produced on it. According to Hartman et al. [7], species that are difficult to propagate are better propagated with softwood cuttings than with hardwood ones. To date, the possibility of vegetative propagation of the ‘GiSelA 5’ rootstock by means of hardwood shoot cuttings [9,10,11,12,13], has been investigated. Experimentally, the root formation levels of hardwood cuttings were found to be higher compared to softwood and semi-hardwood cuttings. The best results were obtained when cuttings were obtained on two dates (the second half of November and early March) at a length of 20 cm [13]. On the other hand, good results (65–80%) in root formation in softwood cuttings of this rootstock have been obtained by other authors [8,14] when using auxins IBA and IAA in the form of an alcoholic solution. Doric et al. [15] recorded the highest percentage of rooting (93%) of the micropropagated ‘GiSelA 6’ rootstock with the use of auxin IBA (1 mg∙L^−1^).

However, the production of synthetic auxins is currently not permitted within the European Union. Hence, research is currently underway into the use of biostimulants in the root formation in shoot cuttings of various plant species [16,17,18,19,20,21]. Among other observations, better root formation in semi-hardwood cuttings of *Chrysanthemum* sp. and softwood cuttings of *Lavandula* × ‘Frills’ after application of the biostimulant RootNectar^®^ (extracts of aloe vera, brown algae (*Ascophyllym nodosum*) and willow bark) was noted [18]. *Begonia semperflorens* cuttings treated with Radifarm^®^ (amino acids, betaines, glycosides, polysaccharides, saponins, organic acids, vitamins and micronutrients) which absorbed nutrients more efficiently, had higher free proline content in roots, fresh and dry plant weight and more leaves and flowers and increased tolerance to low temperatures [19]. In addition, this biostimulant increased the fresh and dry weight of the roots and above-ground parts, as well as the number of leaves and flowers of *Tagetes patula* [20]. A positive effect of the Quik-link^®^ formula (micronutrients, biologically active organic compounds such as plant amino acids and peptides) was observed on the emergence of adventitious roots of *Ocimum basilicum* ‘Genovese’, *Solanum lycopersicum* ‘Washington Cherry’ and *Chrysanthemum indicium* ‘Hollister’ cuttings [21]. Humic substances are also used as root growth biostimulants as an alternative to synthetic preparations, due to their ability to improve nutrient uptake and distribution by plant roots [22]. A significantly higher root formation rate of *Coffea arabica* ‘Topázio’ cuttings was found after the application of humic acids and *Cyperus haspan* extract [23].

Taking these results into account, an experiment was conducted to compare the effects of different synthetic preparations and two biostimulants on the root formation efficiency in shoot softwood cuttings in the ‘GiSelA 5’ rootstock. In addition, the use of two different substrates and the timing of obtaining cuttings was evaluated.

## 2. Materials and Methods

### 2.1. Plant Material and Growth Conditions

Shoots for the preparation of softwood cuttings of the ‘GiSelA 5’ rootstock were obtained from the authors’ own parent plants that were eight years old and free from visual signs of disease. Cuttings 10 cm in length were prepared on two dates: the first and third years in the month of June. Three series of experiments were conducted between 2017 and 2019. The cuttings were rooted in plastic plug trays using two types of substrate. The first was a mixture of coarse river sand and acid peat at a ratio of 1:2, (pH 6.0). The second substrate was TS1—a ready-for-use peat substrate (Klasmann–Deilmann) mixed with perlite at a ratio of 2:1 (pH 6.5). Both substrates were with the addition of PG Mix fertilizer at 1 kg·m^−^³. The softwood cuttings were rooted in a low polytunnel which was located in an unheated greenhouse. The tunnel was equipped with an automatic fogging system, and heating cables covered with a 3 cm layer of coarse sand were installed in the furrow beds where the cuttings were placed. During the root formation in the cuttings, the temperature in the tunnel was kept between 18 and 28 °C and the humidity was between 80 and 85%. Air humidity was recorded using a Hobo measuring device.

The experiment involved six treatments on cuttings with root formation aids and a control (without any aid). The names of the formulations, their concentration and their method of application are shown in Table 1. The experiment consisted of 28 combinations (seven treatments on cuttings, two substrate types and the timing of cuttings collection). Each combination consisted of 30 cuttings, 10 each in triplicate.

A spray treatment was carried out three times at two-week intervals. The first was immediately after the cuttings had been prepared and placed in the substrate. The Bispeed formulation comprises three groups of nitrophenolates: potassium 4-nitrophenolate (potassium para-nitrophenolate) 0.25–0.30% m/m; potassium 2-nitrophenolate (potassium ortho-nitrophenolate) 0.14–0.20% m/m; potassium 5-nitroguaiacolate (potassium 2-methoxy-5-nitrophenolate) 0.07–0.10% m/m. Goteo contains GA142-biologically active extracts from *Ascophyllum nodosum*, 13.0% phosphorus pentoxide (P_2_O_5_) and 5.0% potassium oxide (K_2_O) soluble in water. The concentrations of biostimulants were in line with the manufacturer’s recommendations. In order to protect the seedlings from fungal diseases during the root formation period, preventive spraying treatments were carried out with the fungicides: Amistar 250 SC, Previcur 840 EC and Rovral Aquaflo 500 SC at a concentration of 0.2% each.

### 2.2. Plant and Physiological Parameter Measurements

Physiological processes on the cuttings were measured twice in 2018–2019, two weeks after the start of root formation in the cuttings. The following parameters were measured with a CI-340 aa Handheld Photosynthesis device (CID Bio-Science Inc., Camas, WA, USA): net photosynthetic rate (Pn, μmol CO_2_·m^−2^·s^−1^), transpiration rate (E, μmol H_2_O·m^−2^·s^−1^), stomatal conductance (C, mol H_2_O·m^−2^·s^−1^), intracellular CO_2_ (I CO_2_, mol CO_2_·mol^−1^). The research was conducted at a constant intensity of photosynthetically active radiation (PAR) (1000 µmol·m^−2^·s^−1^) supplied to the plants and at a constant concentration of carbon dioxide (CO_2_) (390 µmol CO_2_·mol^−1^ of air). Healthy, mature leaves were obtained at random for measurements. Eight measurements were obtained for each treatment of shoot cuttings tested, including two for each substrate type and timing of cuttings. The impact of the timing was found to be non-significant so it was omitted from further description of the results.

For each treatment, leaves were obtained, before fall, from eight randomly selected rooted cuttings, two samples from each substrate and timing, in four replicates. The results in Table 2 and Table 3 are presented as mean values for the treatments since no significant effect of substrate or timing was found. The collected leaves were weighed to determine their fresh weight [g] and then scanned, and from the resulting scans, the total leaf blade area [cm^2^] per cutting was calculated in the ‘SKWER’ software. The percentage of rooted cuttings in relation to those placed in the substrate was also determined. At the end of growth, after the cuttings were removed from the substrate, the number of roots was counted and the fresh weight of all cuttings without leaves was weighed.

### 2.3. Statistical Data

Statistical calculations were performed using Statistica 13.1. The Duncan test was used to perform the statistical analysis of the results, with *p* = 0.05. The percentage of rooted cuttings was identified using the arcsine transformation. The parameters of the cuttings were calculated using a three-factor analysis of variance, (preparation, substrate, date). Analyses were performed separately for each year of the experiment. Fresh weight and leaf blade areas, as well as the intensity of physiological processes, were determined also using three-factor analysis of variance for each of the two years separately.

## 3. Results

### 3.1. The Growth Parameters of Softwood Cuttings of ‘GiSela 5’ Rootstocks

The results obtained from the interaction of the three factors, based on their number, are presented in Table 2, Table 3 and Table 4 only as averages for each treatment and for kind of substrate and date of collection of cuttings are listed in the description below. The percentage of rooted leaf cuttings of the rootstock ‘GiSelA 5’ in the first year of the experiment was highest for Rhizopon AA (Table 2). It was lower for Ukorzeniacz AB and the biostimulant Bispeed, then for the two auxins IAA and IBA. The lowest score was recorded for the control combination. The peat–sand substrate proved better (75.9 b) than the second substrate used in the experiment (70.5 a). The earlier date also resulted in a higher percentage of root formation in cuttings (75.4 b) than the later date (71.0 a). In 2018, the highest percentage of rooted cuttings was found for the biostimulant Bispeed and the auxin IBA. It was lower for the biostimulant Goteo and Ukorzeniacz AB. They were followed, in descending order, by Rhizopon AA and auxin IAA. The lowest percentage of root formation in saplings was again obtained for the control. Peat with sand (77.5 b) was a more effective substrate than the second substrate analyzed (73.0 a) and an earlier date (77.0 b) was more effective than a later one (73.6 a).

In the last year of the study, the best results were found for the biostimulant Bispeed and the auxin IBA. This was followed by the Goteo biostimulant, then Ukorzeniacz AB and Rhizopon AA. The worst result was for the control and auxin IAA combination. The higher appropriateness of using a peat–sand substrate (86.0 b) than the other one (81.0 a) and an earlier date for obtaining cuttings (84.3 b) than a later one (82.8 a) was confirmed.

Summarizing the results from the three years, the rooting of the cuttings was best stimulated by Rhizopon AA, followed by Ukorzeniacz AB and auxin IBA, then two biostimulants and auxin IAA. Cuttings that were not treated with any preparation showed the lowest root formation.

The highest number of roots of ‘GiSelA 5’ rootstock cuttings in the first year was found, in the order of the results obtained, for auxin IBA, IAA and Rhizopon AA (Table 3). The other treatments of the cuttings had no differential effect on the parameter under investigation. The mean values for the two types of substrate and timing were not significantly different (substrate: 4.9 a and 4.3 a and timing: 4.8 a and 4.4 a). A similar sequence of results for each treatment of the cuttings was obtained in the following year. Peat with sand (5.6 b) proved to be a better substrate for rooting cuttings than the other substrate (5.0 a). The earlier timing guaranteed a higher number of cutting roots (6.1 b) than the later (4.5 a). In the final year of the study, the order of the effect of the treatments applied to the cuttings on the number of roots was not fully confirmed. Auxin IBA and Rhizopon AA stimulated the growth of the greatest number of roots. Auxin IAA was the next, followed by Bispeed and Ukorzeniacz AB. The lowest root number was confirmed for Goteo and the control. As in the first year of the study, neither the type of substrate nor the timing of cuttings had a significant effect on the number of roots (4.3 a and 3.9 a and 4.1 a for both dates). On average for the three years, cuttings treated with auxin IBA had the highest number of roots, followed by IAA and Rhizopon AA, and the lowest number of roots were found in the other preparations and the control.

The fresh weight of ‘GiSelA 5’ rootstock cuttings in the first year of the experiment (Table 4) was the best for auxins IAA, IBA and Rhizopon AA. In descending order, further results were recorded for Ukorzeniacz AB, followed by Bispeed and the control combination. The lowest fresh weight was observed after the application of Goteo biostimulant. The peat–sand mixture (1.9 b) proved to be a better substrate than the other substrates (1.7 a). The date of collection of the cuttings had no effect on the fresh weight of the cuttings (1.9 a and 1.8 a) respectively. In the second year of the study, as in the first, the best weight of cuttings was found for auxin IAA. Next in order of results were the cuttings stimulated with Rhizopon AA and Ukorzeniacz AB, which had a higher weight than those treated with Bispeed and in the control. The other two factors had no effect on the variation in the weight of the cuttings for the two factors (substrate-2.2 a and 2.3 a, term 2.4 a and 2.5 a) respectively. In the third year, the best fresh weight of cuttings was obtained for Rhizopon AA, Auxin IBA and Ukorzeniacz AB. Cuttings treated with Goteo, Bispeed and the control had the lowest weight. As in the first year of the study, a peat and sand mixture (2.4 b) was a better substrate than the other substrate (2.2 a) and an earlier date for taking cuttings (2.4 b) was better than a later date (2.2 a). The average weight of cuttings for synthetic formulations was better than for biostimulants and controls.

The fresh weight of leaves obtained for the 2018 cuttings (Table 5) was significantly the best for auxin IBA. Next, in descending order were auxin IAA, Rhizopon AA and Ukorzeniacz AB. The lowest fresh weight was found for the control and the two biostimulants. The results of the fresh weight of the leaves of the cuttings in the second year of measurements had a slightly different order regarding the treatments tested. Cuttings treated with auxin IBA and Ukorzeniacz AB had the highest leaf weight, then, in descending order, auxin IAA and Bispeed, followed by Rhizopon AA and the control combination. The lowest value was obtained for the Goteo biostimulant.

The total leaf blade area of the cuttings (Table 5) varied according to the treatment applied. The best result was found with auxin IAA. Two further results were obtained for Rhizopon AA and auxin IBA. This was followed by Ukorzeniacz AB and Goteo. The worst result was achieved for the combination of the control and Bispeed biostimulant. In the following year, the best leaf blade area was obtained for Ukorzeniacz AB, followed by Rhizopon and two auxins, and then two biostimulants. The result for the control combination was the worst.

### 3.2. The Physiological Parameters of Softwood Cuttings of ‘GiSela 5’ Rootstocks

The net photosynthetic rate (Pn) of the 2018 cuttings (Table 6), was highest after the application of Rhizopon AA and auxin IAA. The weakest result for the parameter tested was obtained for the Goteo and Bispeed biostimulants. The results of the other preparations were average and did not differ. The type of substrate did not affect the variation in mean values (substrate with perlite-8.1 a, peat with sand-7.2 a). In the following year of observation, ‘Pn’ for some treatments was lower. The highest value was obtained when auxin IBA was used. The lowest values were obtained for Ukorzeniacz AB and auxin IAA, and average values for the other treatments. The use of a peat-and-sand substrate gave a better result (8.3 b) than the second substrate (5.6 a).

Leaf transpiration coefficient (E) for the year 2018 (Table 6), was highest for auxin IAA, followed by Rhizopon AA. The worst results were obtained for the Goteo and Bispeed formulations. Other combinations produced average results. A mixture of substrate with perlite (1.1 b) proved to be a better substrate than peat with sand (0.7 a). In the following year of the study, the highest value of ‘E’ was found for auxin IBA, with lower values for other formulations. The substrate with perlite again proved to be a better type of substrate (1.3 b) than the second substrate (0.9 a).

Stomatal conductance (C) in the first year of the study (Table 7), was best for auxin IAA, Rhizopon AA, followed by auxin IBA, with the other preparations showing the poorest conductance. No effect of the type of substrate was shown (54.7 a and 49.9 a). In the second year, the highest values of the ‘C’ parameter were obtained for auxin IBA, followed by Rhizopon AA and Goteo, and the lowest value for Ukorzeniacz AB. The mean for the type of substrate used did not differ (34.1 a and 35.1 a).

Intercellular CO_2_ concentration (I CO_2_) in the first year of the study (Table 7), was highest for the Bispeed biostimulant and lowest for Rhizopon AA and the control. The obtained mean values for the type of substrate used were not significantly different (147.2 a and 145.3 a). In the following year, the value for the parameter tested was found to be more than three times higher than in the previous year. The highest ‘I CO_2_’ was recorded for auxin IBA and the lowest for the control. The other treatments showed average results. The use of a substrate with perlite guaranteed a better result (409.1 b) than peat with sand (407.1 a).

## 4. Discussion

Some researchers [7,8] indicate a better root formation efficiency in hardwood cuttings of the ‘GiSelA 5’ rootstock compared to softwood cuttings. In this experiment, a high percentage of root formation in softwood cuttings was obtained on average for all treatments with the highest attainment in 2019 (82.7%). This was a much higher percentage than that found for hardwood cuttings in the aforementioned studies (22% and less than 20%).

In two years of the study, the highest percentage of root formation of cuttings was recorded for Rhizopon AA and in the last year, for the biostimulant Bispeed. The better results found for rooting cuttings in 2019 also found for biostimulators could be due to lower average temperatures outside the greenhouse in the months when the cuttings were rooted (June–August). It could also cause the slower drying of foliar-applied biostimulants, the components of which were taken up in greater amounts by the leaves of the cuttings. Overall, the two tested biostimulants resulted in significantly better root formation in the cuttings than in the control. However, their effects were not as strong as those of synthetic preparations, of which Rhizopon AA and auxin IBA were the best. Similarly, in a study by Gulen et al. [14], auxin IBA, administered in the same way, resulted in a root formation percentage of less than 65%, and the cuttings not treated with auxin did not develop roots at all. This demonstrates the need to use preparations to stimulate root formation in cuttings of this rootstock as stated by other authors [13,14]. Similarly, in another experiment [8], significant differences were found in the percentage of leaf cuttings that were rooted using two forms of auxins, IBA and IAA (2%), versus a control; more than 40%, 20% and 10% of rooted ‘GiSelA 5’ rootstock cuttings were obtained, respectively. The opposite results were obtained by Nečas and Krška [24]: for root formation in softwood cuttings of various *Prunus* rootstocks (‘VVA-1’, ‘AP-1’, ‘Lesiberian’, ‘MY-KL-A’, ‘Ishtara’, ‘PS-1’, ‘MRS 2/5’), using synthetic preparations: Rhizopon AA 2.5% and Racine (2.5% NAA + 2-nitrophenol sodium), no significant variation in root formation percentage was found in cuttings compared to the control. Similarly, Markovski et al. [10] did not show an effect of auxins IBA 2% and NAA 0.2% on root formation in softwood cuttings of various rootstocks (‘Mirabolana’, ‘St. Julian A.’, ‘St. Julian INRA’, ‘St. Julian Orleans’, ‘GF 8/1’, ‘GF 655/2’, ‘Alkavo’, ‘Gisela 4’, ‘Gisela 5’, ‘Weiroot 53’, ‘Weiroot 158’), in relation to the control combination. The results of these studies are not consistent with those obtained in this experiment, where the control combination produced the worst results in all years of the study, of all the treatments used. Similar conclusions were drawn by Sharma and Kumar [12] who were rooting softwood cuttings of ‘Myrocal’, ‘Julior’ and ‘Jaspi’ rootstocks. They obtained a very low root formation percentage with auxin IBA, but it was significantly better than the control (0.0). In the experiment, they conducted shoot cuttings of ‘GiSelA 5’, where the biostimulant Bispeed was used twice in 2017 and 2019, and showed a very high root formation percentage. Szabó et al. [25] also showed the effectiveness of the foliar application of biostimulants based on natural ingredients (Kelpak^®^ 0.2%, Wuxal^®^ Ascofol 0.2%, Pentakeep^®^ -V 0.05%, Yeald Plus^®^ 0.15%, BA 0.2%), on a higher percentage of root formation in softwood cuttings of the rootstock ‘Magyar’ (*Prunus mahaleb* L.).

In the experiment conducted, a trend of the most favourable effect of auxin IBA on the number of roots was noted in all years of the study. Other authors [8,14,26,27] have also found a positive effect of this auxin, reaching about 10 roots, which is more than in the experiment considered (8.1). Kapczyńska et al. [28], demonstrated a positive effect of the biostimulant Goteo at a concentration of 0.1% on the length of roots of the cuttings of *Pennisetum purpureum* ‘Vertigo’ grass, compared to the control. This is not confirmed in the experiment conducted, where the tested biostimulator had a similar effect to the control. This could be due to the different dynamics of biostimulant uptake by perennial tree compared to grass.

The best fresh weight of cuttings, in all years of the study, was obtained for cuttings treated with synthetic auxins. In an analysis of the effect of the biostimulants on the parameter in question, no positive impacts on increasing the weight of the cuttings were found, compared to the control. In addition, the parameters analyzed for the cuttings, i.e., fresh weight and leaf area, made it clear that the biostimulants used had results for these characteristics on par with the control, without stimulating their growth, but only resulting in better performance of the rooted cuttings.

Research is continually being carried out to select the appropriate composition of the root formation substrate for shoot cuttings of various plant species, in order to ensure optimal air and water conditions [29]. Many authors [28,30,31] indicate a higher root formation percentage of shoot cuttings using perlite as a stand-alone substrate or one of the components compared to sand. This was not confirmed in this experiment. A higher root formation percentage of shoot cuttings was achieved using a mixture of peat and sand as a substrate. Other authors [8,32], who rooted softwood shoot cuttings of ‘GiSelA 5’, used a mixture of peat and sand (3:1), or used a mixture of RS II peat substrate with perlite (2:1) [9]. Exadaktylou et al. [12] obtained the highest root formation percentage of ‘GiSelA 5’ hardwood cuttings, using a peat–perlite mixture (1:1), and perlite alone, as a substrate. The various results of over mentioned experiments suggest that the root formation percentage in the ‘GiSelA 5’ rootstock cuttings in question is less dependent on the type of substrate used and more on the type of rooting stimulant used. In the experiment in question, the better root formation percentage in the sand and peat substrate may also have been due to the fact that the substrate dried out more slowly compared to the peat–perlite substrate when using heating cables on which the containers with the cuttings were placed.

Kapczyńska et al. [28] found variation in the number of roots, of cuttings rooted in different substrates. Similar differences for ‘GiSelA 5’ cuttings were noted in this experiment, in favour of the peat with sand. Aghdaei et al. [33] conducting a study on the effect of 11 substrates on the rooting of shoot cuttings of *Solanum muricatum* (Aiton), proved the most favourable effect on the number of roots of the substrates: a mixture of peat with perlite, and peat with sand, which confirms their own observations on the suitability of both these substrates. Moreover, their effect on the results obtained was not very strong.

Another factor which was analyzed that influenced the number and quality of rooted cuttings was the timing of the collection. This is because it determines the amount of naturally occurring auxin IAA in the plant, which can be a decisive factor in identifying the optimum time for the propagation of a given species. However, it also depends on other physiological and environmental factors [29]. Mezey and Leško [9] collected material for ‘GiSelA 5’ cuttings on four dates (monthly from July to October). They received the highest percentage of rooted cuttings for cuttings collected in October, highlighting the impossibility and low profitability of propagating this rootstock by cuttings in earlier months. This was not supported by this experiment. By testing two dates (the first and the third years in the month of June), a high and significantly better root formation percentage of shoot cuttings of ‘GiSelA 5’ was obtained for cuttings collected at the earlier date. The effectiveness of the summer term is confirmed by studies carried out by other authors [32]. By obtaining ‘GiSelA 5’ cuttings in mid-June, they obtained a high percentage of root formation (77.8–98.9%) for mid and top cuttings, respectively. In the authors’ own experiment, mid cuttings were obtained and similar results were obtained depending on the type of root formation stimulant.

The obtained results of the physiological parameters (Pn, E, C, CO_2_) of ‘GiSelA 5’ cuttings varied depending on the root formation stimulants used, and to a lesser extent, depending on the substrate used. Studies conducted to date confirmed that the use of biostimulants increases the photosynthetic rate [33,34,35,36], which was not fully confirmed in this experiment. The 2019 results indicated an improvement in most of the photosynthetic parameters tested (Pn, E, C) after the use of the biostimulants Goteo and Bispeed, compared to the control. However, they had reduced the values of the same parameters in the previous year. In a study by other authors [37] on smoke tree (*Cotinus coggygria* Scop.) shoot cuttings, the application of biostimulants (AlgaminoPlant and Route) significantly increased the photosynthetic rate compared to the control. Furthermore, it sometimes had a more beneficial effect on the parameters tested than synthetic preparations (Rhizopon AA, auxin IBA), which was rarely confirmed in this experiment. Positive effects of biostimulants on the photosynthetic rate of shoot cuttings have also been reported in other-dogwood (*Cornus mas* L.) [38] and ninebark (*Physocarpos opulifolius* L. Maxim) species [39]. The biostimulants increase CO_2_ levels (more than twice forGoteo and three times for Bispeed) only in 2018 compared to the control, while at the same time, the levels of other tested parameters were lower. However, no similar trend was noticed in the following year. The reason for the low photosynthetic rate of the ‘GiselA 5’ rootstock cuttings may have been the incomplete utilization of stored CO_2.._ Similarly, other authors [40] found a positive effect of foliar-applied biostimulants on tomato plant growth, while these preparations did not increase CO_2_ levels. A similar trend was noticed for some treatments in this experiment. In the experiment under consideration, the majority of synthetic preparations showed a greater leaf area of the cuttings, and consequently, a higher photosynthetic rate compared to the biostimulants and the control. However, in studies by other authors [41,42], stronger growth was not always associated with better photosynthetic parameters.

Research into the vegetative propagation of the ‘GiSelA 5’ rootstock should continue, given its high usefulness in starting new intensive sweet cherry orchards. In future studies, attention should be paid to the application of other biostimulants to mother plants, before collecting cuttings, which may increase the level of natural auxins stored in the plant.

## 5. Conclusions

The presented study confirmed the possibility of using biostimulants during root formation of softwood rootstock cuttings of ‘GiSelA 5’. Their impact was comparable to synthetic formulations in terms of the percentage of rooted cuttings. However, it did not result in stronger growth of cuttings compared to the control, which was confirmed with a measurement of the intensity of physiological parameters. For species that are difficult to root, such as ‘GiSelA 5’ rootstock, the use of synthetic auxins in powder and/or alcohol solution may be necessary to achieve economical results. The effect of individual synthetic auxins on the rooting of cuttings was comparable and it is difficult to indicate a specific one. This requires further experimentation to compare different concentrations and forms of application. The type of substrate used was less significant and did not cause much difference in the results obtained. The timing of the collection of cuttings in June reached satisfactory results and should be recommended for the propagation of ‘GiSelA 5’ rootstock.

## Figures and Tables

**Table 1 plants-12-00658-t001:** Used treatments for root formation ‘GiSelA 5’ softwood cuttings.

Treatments	Method of Application
Control	Three-spray treatment with distilled water
Ukorzeniacz AB (0.2% NAA; 0.1% IBA; 0.1 % amid NAA) powder	One treatment and three-spray treatment with distilled water
Rhizopon AA (0.2% IBA) powder	One treatment and three-spray treatment with distilled water
IAA (2 g·L^−1^) the auxins were dissolved in pure ethanol and filed up with water to obtain 1%	Quick-dipped for about 5 s and three-spray treatment with distilled water
IBA (2 g·L^−1^) the auxins were dissolved in pure ethanol and filed up with water to obtain 1%	Quick-dipped for about 5 s and three-spray treatment with distilled water
Goteo 0.2%	Three-spray treatment
Bispeed 0.5%	Three-spray treatment

**Table 2 plants-12-00658-t002:** Percentage of obtained softwood cuttings of GiSelA 5 rootstock depending on the tested treatments in the years 2017–2019.

Treatments	2017	2018	2019	Average for Three Years
Control	59.2 a	60.2 a	72.2 a	63.9
Ukorzeniacz AB	76.9 d	83.4 e	83.7 cd	81.3
Rhizopon AA	85.0 e	85.4 f	82.4 d	84.3
IAA	71.3 c	78.2 d	75.3 b	74.9
IBA	72.7 c	75.1 c	91.0 e	79.6
Goteo	67.2 b	77.2 d	84.2 d	76.2
Bispeed	78.0 d	64.4 b	92.0 e	78.1

Data followed by the same letters do not differ significantly at *p* = 0.05 for each parameter according to Duncan’s test.

**Table 3 plants-12-00658-t003:** Roots number of softwood cuttings of GiSelA 5 rootstock depending on the tested treatments in the years 2017–2019.

Treatments	2017	2018	2019	Average for Three Years
Control	1.5 b	1.9 a	2.1 a	1.8
Ukorzeniacz AB	2.0 c	2.4 cd	2.3 bc	2.2
Rhizopon AA	2.1 cd	2.5 cd	2.6 d	2.4
IAA	2.3 d	2.6 d	2.2 ab	2.4
IBA	2.1 cd	2.3 bc	2.5 cd	2.3
Goteo	1.3 a	2.1 ab	2.0 a	1.8
Bispeed	1.6 b	2.0 a	2.1 ab	1.9

Data followed by the same letters do not differ significantly at *p* = 0.05 for each parameter according to Duncan’s test.

**Table 4 plants-12-00658-t004:** Fresh mass (g) of softwood cuttings of GiSelA 5 rootstock depending on the tested treatments in the years 2017–2019.

Treatments	2017	2018	2019	Average for Three Years
Control	3.5 a	4.1 a	2.4 a	3.3
Ukorzeniacz AB	3.7 a	4.1 a	3.7 b	3.8
Rhizopon AA	4.2 b	6.1 b	5.6 d	5.3
IAA	6.3 c	6.4 b	4.7 c	5.8
IBA	8.0 d	7.6 c	5.7 d	7.1
Goteo	2.9 a	4.4 a	3.0 a	3.4
Bispeed	3.1 a	4.7 a	3.7 b	3.8

Data followed by the same letters do not differ significantly at *p* = 0.05 for each parameter according to Duncan’s test.

**Table 5 plants-12-00658-t005:** Fresh mass (FM) and leaf blade area (LBA) of softwood cuttings of ‘GiSelA 5’ rootstock depending on the tested treatments in the years 2018–2019.

Treatments	2018 FM (g)	2019 FM (g)	2018 LBA (cm^2^)	2019 LBA (cm^2^)
Control	0.28 a	0.95 ab	45.04 a	34.37 a
Ukorzeniacz AB	0.31 a–c	1.10 c	50.38 bc	40.79 d
Rhizopon AA	0.32 bc	0.95 ab	52.55 c	39.42 cd
IAA	0.33 c	0.98 b	55.98 d	38.09 b-d
IBA	0.37 d	1.12 c	51.58 c	37.81 b-d
Goteo	0.30 ab	0.92 a	47.19 ab	35.94 ab
Bispeed	0.28 a	0.97 b	45.05 a	37.11 a–c

Data followed by the same letters do not differ significantly at *p* = 0.05 for each parameter according to Duncan’s test.

**Table 6 plants-12-00658-t006:** Net leaf photosynthesis level (Pn, µmol CO_2_·m^−2^·s^−1^) and leaf transpiration coefficient (E, µmol H_2_O·m^−2^·s^−1^) of softwood cuttings ‘GiSelA 5’ depending on the tested treatments in the years 2018–2019.

Treatments	2018 Pn	2019 Pn	2018 E	2019 E
Control	5.65 b	5.87 ab	0.73 b	0.70 a
Ukorzeniacz AB	4.91 b	3.60 a	0.60 b	0.70 a
Rhizopon AA	15.37 c	8.60 bc	1.42 c	1.28 a
IAA	15.67 c	4.03 a	1.79 d	1.07 a
IBA	6.03 b	10.18 c	0.84 b	2.37 b
Goteo	1.79 a	8.23 bc	0.29 a	1.01 a
Bispeed	1.00 a	6.88 a–c	0.22 a	0.94 a

Data followed by the same letters do not differ significantly at *p* = 0.05 for each parameter according to Duncan’s test.

**Table 7 plants-12-00658-t007:** Stomatal conductivity (C, mol H_2_O·m^−2^·s^−1^) and internal concentration of carbon dioxide (I_CO_2,_ mol CO_2_·mol^−1^) of softwood cuttings ‘GiSelA 5’ depending on the tested treatments in the years 2018–2019.

Treatments	2018 (C)	2019 (C)	2018 (I CO_2_)	2019 (I CO_2_)
Control	31.07 ab	23.86 ab	80.62 a	400.95 b
Ukorzeniacz AB	27.03 ab	16.48 a	113.38 ab	412.44 ef
Rhizopon AA	76.56 c	41.25 b	56.25 a	409.03 cd
IAA	130.38 d	27.64 ab	165.39 bc	410.91 de
IBA	41.27 b	70.75 c	128.97 ab	413.86 f
Goteo	15.39 ab	36.66 b	204.23 cd	405. 67 b
Bispeed	11.07 a	30.61 ab	247.03 d	408.12 bc

Data followed by the same letters do not differ significantly at *p* = 0.05 for each parameter according to Duncan’s test.

## Data Availability

Data sharing not applicable.

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
