# Peer review of "Assessment of the Effect of Treating ‘GiSelA 5’ Softwood Cuttings with Biostimulants and Synthetic Auxin on Their Root Formation and Some of Their Physiological Parameters"

_plants, 2023, doi:10.3390/plants12030658_

Round 1

Reviewer 1 Report

Text need to be revised due the presence of a several typos. Table 4 do not indicate a unit of measure, affecting data presentation and interpretation. All other aspects are correct.

Author Response

Thank you for reviewing the work.

Text reviewed and typo fixes made. The unit of measurement is also given

in Table 4.

Reviewer 2 Report

Line 53: correct the root formation

You measured the number of the roots but I would like also to measure the length and the fresh or dry weight of them

Author Response

Thank you for reviewing the work.

Other parameters of cuttings growth, such as the sum of the length of the roots and the fresh and dry weight of the roots, were not included.

The first parameter was evaluated but was not included in the publication because it fully reflected the dependencies that resulted in relation to the number of roots. The multiplicity of combinations (28) does not allow to include all the results of the examined features in a specific number of pages of the publication.

The fresh and dry weight of the roots was not determined because all rooted cuttings were used to establish further experiments in the nursery in the field.

The text has been corrected, especially for many typos.

Reviewer 3 Report

Dear author,

compliments on a very interesting study. It has some moderate issues to be solved. 

Line 53: Please correct roo to root formation.

Lines 50 - 69 do not seem to best align with the topic. Herbaceous flowering plants have different plant hormone levels and requirements than perennial deciduous trees. Instead of comparing with herbaceous plants, or in addition to those already present you might consider the following publications:

https://www.notulaebotanicae.ro/index.php/nbha/article/view/9671

https://journals.tubitak.gov.tr/biology/vol39/iss4/8/

https://link.springer.com/article/10.1007/s11240-012-0197-7

Also, in the introduction section, please highlight the significance of Gisela 5, its popularity, advantages, and disadvantages of its usage. 

Gisela 5 enabled most of the spindle growth forms worldwide, providing agro techniques and  harvest from the ground.

M&M section:

how do you know that the plants were free of disease? From where did the plant material initially derive?

Compliments on table 1, it is very beneficial. 

Line118: Healthy leaves were taken at random for measurements. Mature leaves or newly formed ones? 

Tables 2 and 3 belong to the results section. 

Line 130: and the fresh weight of all cuttings without leaves was weighed... why not the whole plants during the vegetation?

Lines 197 and 198 avoid repeating the same words. Please use synonyms. 

Table 4 title has some uneven formatting. 

Line 306: In two years of the study, the highest percentage of root formation of cuttings was recorded for Rhizopon AA and in one year, for the biostimulant Bispeed. 

I noticed the frequent differences among the years, could you please explain the variation in meteorological parameters during the investigated period? Were there any extremes that disturbed the controlled environment? 

Line 342:  This is not confirmed in the experiment conducted, where the biostimulator tested had a similar effect to the control.

... which was expected due to the different nature of the grass and the perennial tree. 

Please amend the conclusion section with precise information so it can be useful on a stand-alone basis. 

Author Response

Almost all of the reviewer's suggestions have been taken into account.

Lines 50-69 have not been omitted because they refer to the impact of biostimulators which have so far been used mainly on annual plants. The publications indicated by the reviewer concern the influence of synthetic auxins on in vitro propagation (one of them was added).

The fresh weight of the cuttings without leaves was weighed as the fresh weight of the leaves was weighed separately. Also not during the rooting of the cuttings, but only after the end of growth, because the cuttings were used for further experiments in the ground nursery, and weighing them during the growing season without the substrate would result in their loss.

Thank you very much for the other valuable suggestions.